# Single-photon emission from two-dimensional perovskites channeled through low-energy edge states

Gunwoo Na [1,8], Jee Yung Park [2,3,4,8], Jae-Pil So [5], Peijun Guo [3,4], Letian Dou [2,6,7] ✉ & Hong-Gyu Park [1] ✉

Two-dimensional perovskites, an emerging family of van der Waals materials with an innate quantum-well structure, exhibit remarkable optical characteristics and compositional tunability. However, their potential as a source for quantum emitters remains unexplored. Here, we investigate low-energy edge states unique to two-dimensional perovskites using comprehensive photoluminescence imaging and spectroscopy, to further leverage them towards uncovering quantum emitters. In contrast to above-bandgap excitation, spatially localized single-photon emitters are realized through sub-bandgap energy excitation at the edges in exfoliated single crystals. Specifically, the sub-bandgap excitation facilitates efficient access to deeply confined states within the bandgap, as charge carriers can channel through low-energy edge states while inhibiting competing processes. Furthermore, we demonstrate single-photon emitters from artificially created edges via strain as well as from two-dimensional perovskites with different quantum-well thicknesses. Our findings provide promising opportunities in the development of two-dimensional quantum emitters, where two-dimensional perovskites may provide added functionalities with versatility in material design, fabrication, and scalability.

Efficient, stable, and scalable single-photon emission from solid-state quantum emitters is important for the advancement of next-generation quantum cryptography and information technology[1,2]. For example, vacancy-dopant defects in diamond and self-assembled semiconductor quantum dots have been extensively investigated as single-photon emitters[3–8]. However, the limitations on defect growth in targeted locations and the integration with high-quality cavities make their practical implementation challenging. Van der Waals two-dimensional (2D) materials, including transition-metal dichalcogenides (TMDs) and hexagonal boron nitride (hBN), are also considered viable platforms for single-photon generation due to their

advantageous photon manipulation properties[9–17]. Nonetheless, achieving precise and reproducible control of quantum emitters in 2D van der Waals materials, along with the scalable production of high-quality monolayers for effective integration into photonic circuits, remains a significant challenge[18,19]. It is thus essential to explore a diverse set of prospective 2D material candidates and to have a deeper understanding of their emission characteristics to develop optimal single-photon applications.

2D hybrid halide perovskites have recently garnered considerable interest as an emerging family of 2D materials due to their inherent quantum-well structure and exceptional optoelectronic properties[20,21].

[1]Department of Physics and Astronomy, and Institute of Applied Physics, Seoul National University, Seoul, Republic of Korea. [2]Davidson School of Chemical Engineering, Purdue University, West Lafayette, IN, USA. [3]Department of Chemical and Environmental Engineering, Yale University, New Haven, CT, USA. [4]Energy Sciences Institute, Yale University, West Haven, CT, USA. [5]Department of Physics, Soongsil University, Seoul, Republic of Korea. [6]Department of Chemistry, Purdue University, West Lafayette, IN, USA. [7]Department of Chemistry, Emory University, Atlanta, GA, USA. [8]These authors contributed equally: Gunwoo Na, Jee Yung Park. ✉e-mail: letian.dou@emory.edu; hgpark@snu.ac.kr

In contrast to conventional 2D materials, 2D perovskites do not require reduction to a monolayer to exhibit strong quantum confinement effect, due to their alternating organic-inorganic hybrid superlattice-like architecture[22]. These materials possess a general chemical formula of $(L)_2(A)_{n-1}M_nX_{3n+1}$, where L represents a long-chain organic cation, A is a small organic cation (e.g., methylammonium [MA$^+$]), M is a metal cation (such as lead or tin), and X is a halide anion; the integer $n$ defines the quantum-well thickness per unit cell. Variations in each component present a viable approach to tailor specific properties[23–25]. Due to their high tunability in both organic and inorganic layers, multi-faceted design principles, including molecular engineering, lattice dynamics control, and quantum confinement modulation can offer material optimization necessary for reliable single-photon generation. Moreover, 2D perovskites are of low-cost and easily solution processable at scale while maintaining the advantages of van der Waals materials, such as efficient device integration as well as quantum-state manipulation through defect control and strain engineering. Thus, achieving single-photon emission from 2D perovskites will present a potential foundation for scalable and reliable quantum emitters while inspiring new design strategies that extend beyond our current understanding of these materials.

In this work, we explore low-energy edge states that are unique to 2D perovskites with $n \geq 2$ by using detailed photoluminescence (PL) imaging and spectroscopy, aiming to uncover their potential as alternative quantum emitters. Our investigation indicates that spatially confined single-photon emitters can be realized by stimulating the edges in exfoliated single crystals using sub-bandgap energy optical excitation. Through extensive photoluminescence excitation (PLE) analysis, we uncover that sub-bandgap optical excitation effectively allows charge carriers to access deeply confined states within the bandgap through low-energy edge states while suppressing other recombination channels, including band-edge emission or shallow defects. In addition, emitter-site control is achieved using nanostructures that induce edge-like effects even within the interior region of exfoliated sheets. We further develop single-photon emitters from 2D perovskites with varying quantum-well thicknesses, offering insight into their tunability.

## Results

Distinct optoelectronic properties have been observed in 2D perovskites, with differences noted between the interior and the edges of exfoliated single crystals or grains[26–30]. In particular, edge states exhibit significant differences in PL spectra and lifetimes, effective charge-carrier transport, and exciton dissociation in 2D perovskites with two or more inorganic layers ($n \geq 2$), which are attributed to edge-termination-induced reconstruction or termination-dependent electrostatics[29,31]. Therefore, we investigate the edge regions of exfoliated single-crystal sheets of 2D perovskites with varying inorganic layer thicknesses using diverse pump wavelengths (Fig. 1a). We examine three types of 2D perovskites: $(BA)_2PbI_4$ (BA $n = 1$), $(BA)_2(MA)Pb_2I_7$ (BA $n = 2$), and $(BA)_2(MA)_2Pb_3I_{10}$ (BA $n = 3$), where BA stands for butylammonium, to identify the unique optical characteristics of each compound and uncover potential quantum emitters. Figure 1b illustrates the representative alternating organic-inorganic hybrid crystal structure of BA $n = 2$ together with its corresponding quantum-well band structure. Phase-pure bulk single crystals were obtained via a slow cooling crystallization method and subsequently vacuum-dried for mechanical exfoliation into thin sheets (see "Methods"). Supplementary Fig. 1 shows the PXRD results for each single crystal batch, confirming their high compositional purity.

Figure 1c shows a top-view scanning electron microscope (SEM) image of an exfoliated single-crystal sheet of BA $n = 2$, distinctly illustrating the terrace-like morphology of the edge region, with the inset providing a closer view of the edge from an alternative viewing angle. In addition, a notable difference in lattice characteristics is observed between the interior and edge regions of a thin exfoliated sheet, as shown in the high-angle annular dark-field scanning transmission electron microscopy (HAADF STEM) image (Fig. 1d, left). The interior region exhibits clear lattice planes with an interplanar spacing of 0.315 nm corresponding to lattice spacing along the [202] direction, as confirmed by the fast Fourier transform (FFT) pattern (Fig. 1d, right). In contrast, the edge region has a less defined structure, with the FFT pattern highlighting the structural inhomogeneity between the interior and edge regions. Although the extent of the edge is on the order of a few nanometers as shown in the STEM image, the single-crystal sheets investigated in this work hereon were carefully controlled during exfoliation to exhibit a continuous terrace-like edge on the order of micrometers as shown in the inset of Fig. 1c, enabling selective examination of the edges to ascertain their distinct properties.

Comprehensive PL imaging and characterization of 2D perovskites were performed to examine the relationship between emission and pump wavelengths, as well as to locate spatially confined emitters. Figure 2a shows the optical microscope image of an exfoliated single-crystal sheet of BA $n = 2$, which was used for PL measurements in its interior and edge regions at 4 K. The PL spectra in Fig. 2b indicate that the interior region is characterized by the narrow emission at a wavelength of around 580 nm, corresponding to the emission from band-edge exciton recombination of BA $n = 2$. In contrast, the edge region displays a pronounced dominance of lower-energy broad emission. Spatial PL mapping was performed to further clarify that these distinct features are from the edge. First, the narrow emission component was acquired using a 600 nm short-pass filter in conjunction with above-bandgap excitation from a 532 nm continuous-wave (CW) laser (Fig. 2c). No substantial difference in PL intensity was observed between the interior and edge regions. Next, we performed spatial PL mapping of the broad emission component using a 650 nm long-pass filter at various pump wavelengths (Fig. 2d–g). Figure 2d shows the PL mapping of broad emission under above-bandgap excitation at 532 nm, which is the same as that of Fig. 2c; however, the PL intensity shows increased intensity in the edge region than in Fig. 2c. To investigate this feature more clearly, we further increased the pump wavelength. Near-bandgap excitation at 580 nm increased the relative intensity of the edge region in the PL map of broad emission (Fig. 2e), whereas sub-bandgap excitation at 600 nm led to almost completely localized emission in the edge region (Fig. 2f). Finally, spatial PL mapping under even lower-energy sub-bandgap excitation at 633 nm yielded strongly localized emitters at the edges (Fig. 2g).

Conventional above-bandgap excitations are incapable of producing spatially confined emitters in 2D perovskites. Lower-energy excitation leads to increasingly localized PL emissions in the edge region, ultimately yielding highly confined emissive spots. These features observed at 4 K are also present at room temperature (Supplementary Fig. 2), indicating that lowering the temperature does not substantially alter the characteristics of the edge states. In addition, BA $n = 3$ showed similar results to BA $n = 2$, revealing pronounced broad PL emission characteristics in the edges and localized emitters with sub-bandgap excitations (Supplementary Fig. 3). However, exfoliated sheets of BA $n = 1$ exhibit both narrow and broad emission without clear spatial confinement features, regardless of changes in the pump wavelength (Supplementary Fig. 4); spatially localized emitters were not identified even at lower-energy excitations. Broad emissions from $n = 1$ 2D perovskites are typically attributed to defects or self-trapped excitons, which cannot be directly excited and must transition from band-edge excitons[32,33]. Our results are consistent with prior observations that $n = 1$ 2D perovskites do not exhibit edge-state PL, as opposed to $n = 2$ and 3[34]. Thus, we focus on BA $n = 2$ and BA $n = 3$ for further development into single-photon emitters hereon in this work.

We then performed systematic optical experiments to analyze the highly confined emissive spots observed at the edges in BA $n = 2$ under

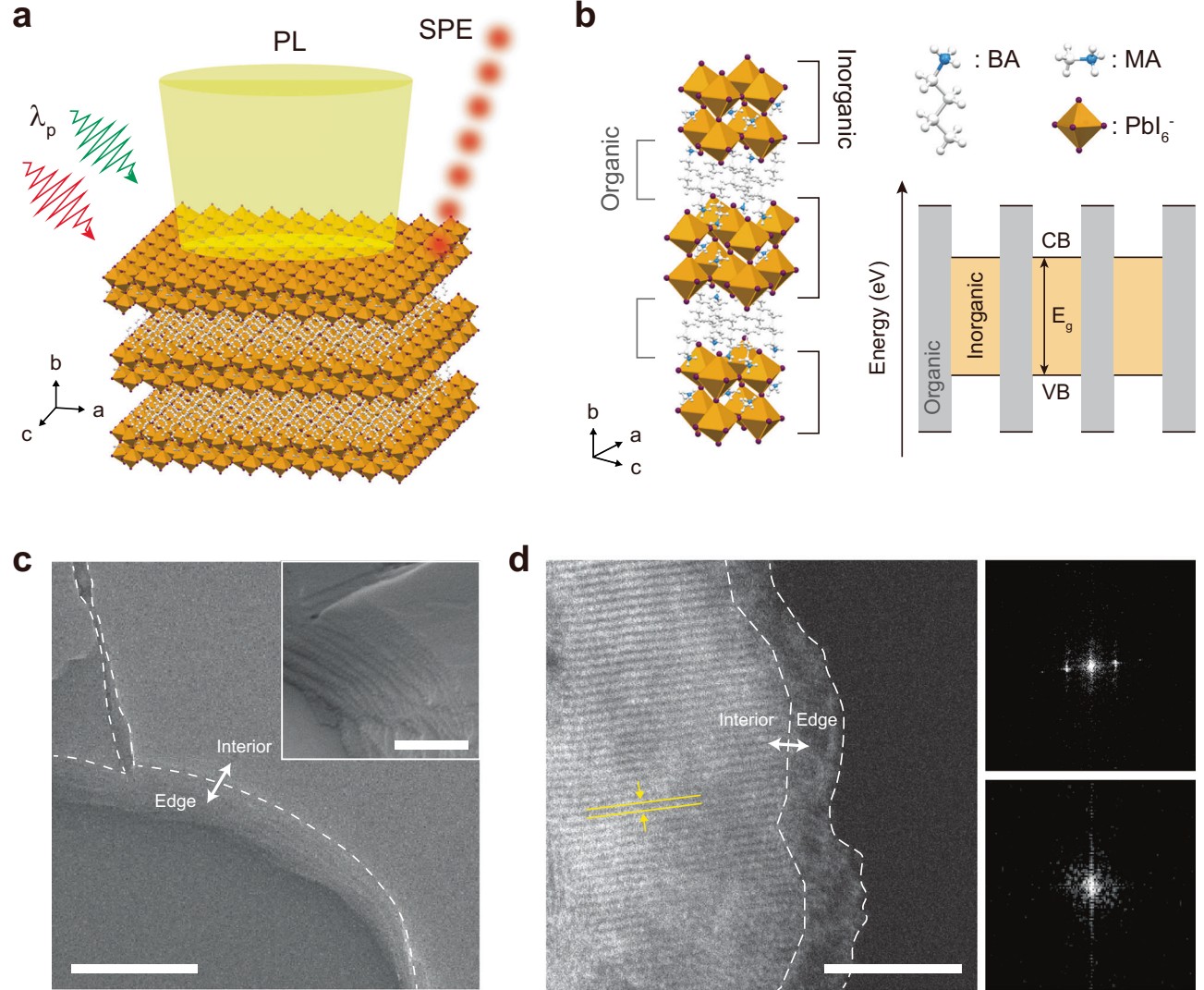

**Fig. 1 | Analysis of the interior and edge structures of 2D perovskites.**
**a** Schematic illustration of wavelength-tunable excitation ($\lambda_p$) in the interior and edge regions of an exfoliated 2D perovskite single crystal for optical characterization. **b** Schematics of BA $n = 2$ single-crystal structure (left) and the corresponding quantum-well band structure (right). CB, VB, and $E_g$ indicate conduction band, valence band, and bandgap, respectively. **c** Top-view SEM image of an exfoliated BA $n = 2$ single-crystal sheet. Scale bar, 2 μm. The inset shows a higher-magnification tilted image of the edge at 70-degree angle. Scale bar, 500 nm. **d** HAADF STEM image of an exfoliated BA $n = 2$ sheet (left), highlighting the clear difference in phase between the interior and edge regions. The yellow solid lines represent the lattice plane. Scale bar, 5 nm. Right, FFT patterns of the interior (top) and edge (bottom).

sub-bandgap excitation to identify evidence of photon anti-bunching. Figure 3a shows an optical microscope image of the exfoliated BA $n = 2$ single-crystal sheet used in the measurements. PL emission was measured at 4 K using a 650 nm long-pass filter in conjunction with sub-bandgap excitation from a 633 nm CW laser (see "Methods"). The exfoliated sheets were initially scanned to search for bright localized spots, and following the screening process, a small area containing several bright emitters was selected as a zone of interest for further investigation (Fig. 3b).

Each local bright spot in Fig. 3b, indicated by the dashed circles labeled 1, 2, and 3, was examined for single-photon emission. PL measurements of the emitters at spots 1 and 2 (white dashed circles) exhibited multiple sharp peaks emitted simultaneously, potentially representing different confined emitters adjacent to each other. However, the spectral proximity and overlap of the peaks result in a lack of unambiguous evidence for clear photon anti-bunching behavior (Fig. 3c). On the other hand, the emitter at spot 3 (red dashed circle) showed a fully isolated single peak at a wavelength of 683.0 nm with a full width at half maximum (FWHM) of 0.65 nm (Fig. 3d). The

second-order correlation function $g^{(2)}(\tau)$ was measured using a Hanbury Brown and Twiss (HBT) setup (see "Methods"). The $g^{(2)}(0)$ value for the peak of spot 3 was $0.397 \pm 0.109$ (Fig. 3e), which clearly indicates photon anti-bunching behavior. The measurement of integrated PL intensity as a function of the pump power showed a saturation emission intensity of 44,171 counts/s at a power of 2.8 μW (Fig. 3e, inset). In addition, the time-resolved PL (TRPL) measurement revealed the decay time of the emitter in Fig. 3d to be $870 \pm 3.73$ ps (Fig. 3f).

Taken together, the results in Fig. 3 indicate that single-photon emission is distinctly observable within localized bright spots located at the edges of exfoliated sheets of BA $n = 2$. The rapid decay time of 2D perovskites, in contrast to TMD monolayers (about a few ns)[35,36] and both boron- and carbon-related vacancies in hBN (also about a few ns)[37,38], can directly contribute to the enhanced rate of quantum emission. A single-photon emitter exhibiting similar properties was also observed in BA $n = 3$ exfoliated sheets under sub-bandgap excitation, yielding a $g^{(2)}(0)$ value of $0.278 \pm 0.103$ (Supplementary Fig. 5). BA $n = 3$ showed a red-shifted single-photon emission peak at 743.4 nm compared to BA $n = 2$ (683.0 nm) due to the smaller bandgap in BA

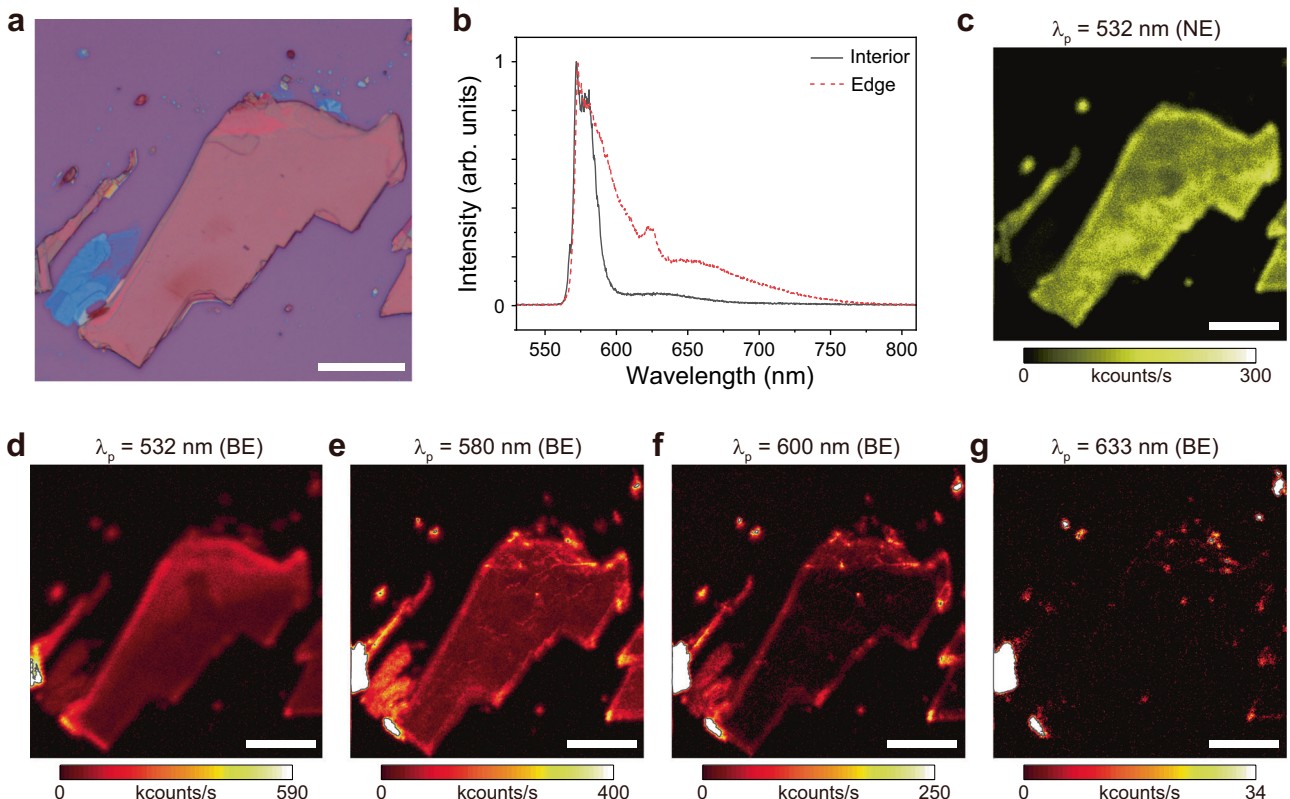

**Fig. 2 | Spatial mapping of PL intensity from BA $n = 2$ at various pump wavelengths. a** Optical microscope image of an exfoliated single crystal BA $n = 2$ used for measurements. Scale bar, 20 μm. **b** PL spectra exhibiting narrow emission from the interior (solid black) and broad emission from the edge (dotted red). **c** Spatial PL map of the narrow emission component (NE) at the pump wavelength ($λ_p$) of 532 nm, using a 600 nm short-pass filter. Scale bar, 20 μm. **d–g** Spatial PL maps of the broad emission component (BE) at the pump wavelength ($λ_p$) of 532 nm (**d**), 580 nm (**e**), 600 nm (**f**), and 633 nm (**g**), using a 650 nm long-pass filter. Scale bar, 20 μm. All measurements are conducted at 4 K.

$n = 3$ with a thicker inorganic layer. In addition, the PL lifetime of BA $n = 3$ was measured to be longer ($τ = 1.729 ± 0.032$ ns) than that of BA $n = 2$ (Supplementary Fig. 6). Overall, these findings establish a unique approach that leverages sub-bandgap, low-energy excitations to generate single-photon emitters in 2D perovskites.

Next, we performed sub-bandgap pulsed excitation measurements on BA $n = 2$ at 4 K. Compared with CW excitation, pulsed excitation suppresses residual background emission, allowing the intrinsic single-photon statistics of the emitter to be measured more reliably and yielding improved $g^{(2)}(0)$ values. Figure 4a shows an optical microscope image of the exfoliated BA $n = 2$ single-crystal sheet used for these measurements. Similar to Figs. 2 and 3, spatial PL mapping with a 650 nm long-pass filter revealed increased intensity in the edge region under near-bandgap excitation ($λ_p = 600$ nm), whereas sub-bandgap excitation ($λ_p = 633$ nm) led to localized bright spots at the edges (Fig. 4b, c). We selected three representative spots (1–3) and observed isolated narrow emission lines with FWHM of -0.65–1.31 nm (Fig. 4d). HBT measurements under pulsed excitation yielded clear photon anti-bunching for all three spots, with $g^{(2)}(0) = 0.229 ± 0.041$ (spot 1), $0.326 ± 0.054$ (spot 2), and $0.345 ± 0.059$ (spot 3) (Fig. 4e). The measurement of the integrated PL intensity as a function of the pump power exhibited saturation behavior, with a saturation emission intensity of 67,213 counts/s at a pump power of 22.2 μW (Fig. 4f).

We also measured power-dependent $g^{(2)}(τ)$ on the same emitter (spot 1 in Fig. 4c) and found that $g^{(2)}(0)$ improves at a lower pump power ($g^{(2)}(0)$ - 0.20 at 20 μW and -0.23 at 40 μW) (Supplementary Fig. 7). This result supports the fact that the observed anti-bunching is dominated by a single emitter, rather than being an artifact arising from background noise. While sub-bandgap excitation yielded single-photon emitters at the edges, we rarely observed localized bright spots

in the interior that exhibited anti-bunching behavior, with $g^{(2)}(0) = 0.471 ± 0.047$ (Supplementary Fig. 8). However, the emission-wavelength stability of these interior point-defect single-photon emitters is distinct from that of edge-state emitters. For quantitative evaluation, we performed time-resolved spectral tracking for both interior point-defect and natural edge-state single-photon emitters (Supplementary Fig. 9). The measurements indicate that edge-state single-photon emitters are less prone to bleaching and exhibit better spectral stability than point-defect single-photon emitters.

To gain deeper insight into three distinct emission characteristics of 2D perovskites, including narrow, broad, and single-photon emissions, we performed photoluminescence excitation (PLE) spectroscopy and TRPL experiments on the edge region of the sample. The PLE map of the BA $n = 2$ exfoliated sheet was obtained by first locating a spot with a confined emitter in the edge region and acquiring PL data from that spot by continuously decreasing the excitation energy from above (528 nm) to below (630 nm) the bandgap (Fig. 5a). The measurement results reveal the following key features. First, the broad emission in the edge region is clearly observed for pump wavelengths ranging from above-bandgap to below-bandgap. This indicates the presence of an energy state at the edge that lies below the conduction band, with the emission from this transition contributing to the broad emission. Second, the broad emission substantially decreases at the pump wavelengths near 590 nm, which corresponds to the bandgap energy of -2.1 eV at the edge. The transition from the edge state is weakened by excitation at an energy corresponding to the bandgap. This phenomenon occurs as a result of resonant pumping, which enables efficient excitation to the conduction band edge[26,39]. Third, a sharp peak is observed at 708.5 nm, with a significant increase in intensity when the pump wavelength changes from above-bandgap to

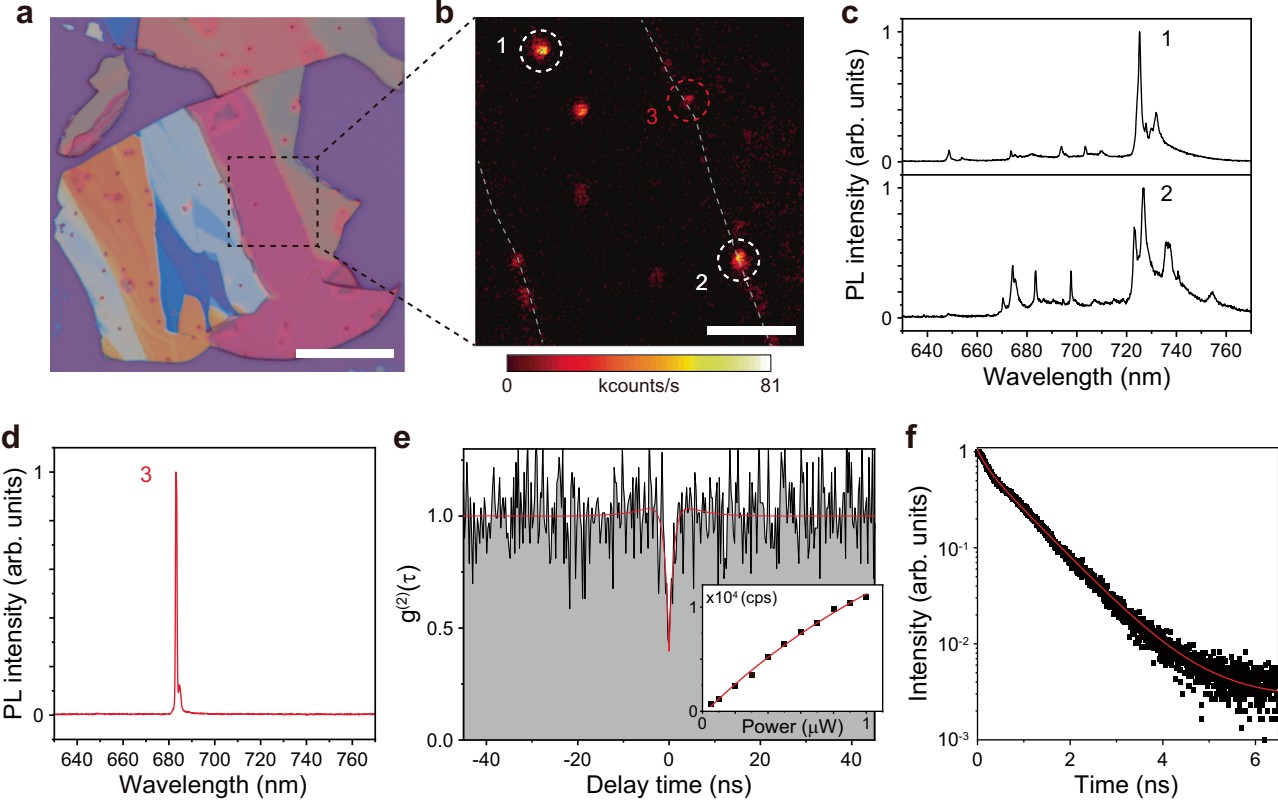

**Fig. 3 | Single-photon emitter located at the edge in BA *n* = 2. a** Optical microscope image of an exfoliated BA *n* = 2 single-crystal sheet on a SiO₂/Si substrate. Scale bar, 20 μm. **b** Spatial PL map of the black dashed area in (**a**). The edges are represented by the gray dashed lines. PL intensity is measured at 4 K under a pump wavelength of 633 nm with a 650 nm long-pass filter. Three localized bright spots labeled 1, 2, and 3 are analyzed. Scale bar, 5 μm. **c** PL spectra for spots 1 and 2 in (**b**) (white dashed circles). **d** PL spectrum for spot 3 in (**b**) (red dashed circle), exhibiting a single sharp peak at 683.0 nm with a full width at half maximum (FWHM) of 0.65 nm. **e** Measured second-order correlation function $g^{(2)}(\tau)$ using the HBT setup. The value of $g^{(2)}(0)$ is 0.397 ± 0.109. The inset shows the light in-light out curve, exhibiting a saturation power of 2.8 μW and a saturation emission intensity of 44,171 counts/s. **f** Time-resolved PL measurement of the single-photon emitter in (**d**, **e**).

near-bandgap and subsequently to sub-bandgap excitation. This feature is particularly evident in Fig. 5b. To verify the single-photon characteristic of the peak, we assessed the second-order correlation function, $g^{(2)}(\tau)$. The fitted value of $g^{(2)}(0)$ was determined to be 0.376 ± 0.167, indicating photon anti-bunching (Fig. 5c). Furthermore, we conducted temperature-dependent PL measurements on the sharp peak, which decreases with increasing temperature and disappears at temperatures over 60 K (Supplementary Fig. 10).

In addition, TRPL experiments were conducted to examine the changes in decay characteristics at different pump wavelengths, using a band-pass filter to isolate the single-photon emission peak (see "Methods"). The TRPL data were analyzed using a biexponential function: $I(t) = A_1 e^{-t/\tau_1} + A_2 e^{-t/\tau_2}$, where $\tau_1$ represents the radiative decay time and $\tau_2$ represents the non-radiative decay time. We measured $\tau_1 = 3.18 \pm 0.14$ ns and $2.59 \pm 0.08$ ns, and $\tau_2 = 0.40 \pm 0.004$ ns and $0.33 \pm 0.002$ ns for the pump wavelengths of 550 nm and 633 nm, respectively (Fig. 5d). We note that $\tau_1$ and $\tau_2$ showed shorter decay times with sub-bandgap excitation at 633 nm. The decrease in both decay times signifies that sub-bandgap excitation activates a channel that allows excitons to be rapidly directed into a mid-gap state and undergo an accelerated recombination process, thereby enhancing the probability and rate of single-photon emission. Furthermore, we performed temperature-dependent TRPL measurements on an edge-state emitter in BA *n* = 2 at 4, 10, 20, 40, and 60 K, and analyzed the spectral characteristics, including the emission linewidth (FWHM) and the integrated PL intensity (Supplementary Fig. 11). We also extracted the temperature dependence of the PL lifetime. A key feature of these

measurements is that the practical temperature limit is primarily set by phonon-driven loss of spectral selectivity together with activated competition from nonradiative/escape channels, rather than a single mechanism alone.

Based on the results of the PLE and TRPL experiments, the model depicted in Fig. 5e elucidates the series of transitions that charge carriers can possibly experience in different regions of 2D perovskite sheets under different energy excitations. First, under above-bandgap excitation, the interior region exhibits a narrow emission characteristic aligned with the bandgap energy (Fig. 5e, left). In the edge region, three distinct channels exist due to the edge state located below the conduction band (Fig. 5e, middle): narrow emission from band-edge recombination, broad emission caused by the transition from the edge state to the valence band, and nonradiative decay through shallow traps[21,40,41] situated just below the conduction band. In contrast, sub-bandgap excitation creates a unique charge carrier transition scenario at specific locations of the edges where mid-gap defect states exist (Fig. 5e, right). Excitons directly excited to the edge state are effectively channeled towards a single mid-gap defect state, enhancing radiative recombination and promoting single-photon emission while other recombination is suppressed, as shown in the PLE measurement in Fig. 5a. We note that the edge state can provide a lower hurdle for excitons to efficiently transition to deeper mid-gap states, likely due to their closer energy levels.

Next, to artificially induce edge-like effects within the interior regions of 2D perovskites, an exfoliated BA *n* = 2 single-crystal sheet was transferred onto a SiO₂/Si substrate including patterned

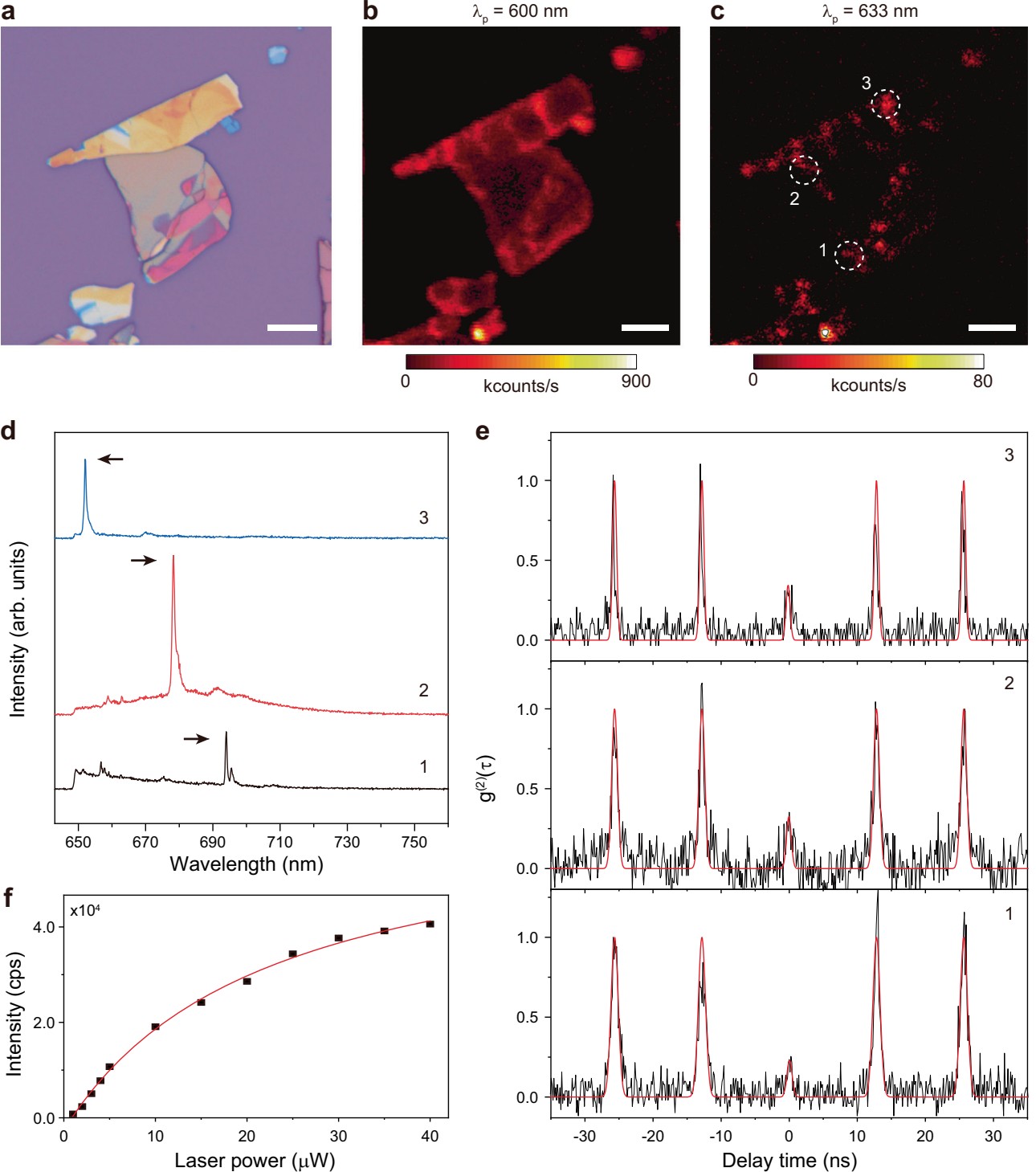

**Fig. 4 | Sub-bandgap pulsed-excitation measurements in BA $n = 2$. a** Optical microscope image of an exfoliated BA $n = 2$ single-crystal sheet on a $SiO_2$/Si substrate. Scale bar, 10 μm. **b, c** Spatial PL maps measured at 4 K using a 650 nm long-pass filter under pulsed excitation at $\lambda_p = 600$ nm (**b**) and 633 nm (**c**). Three localized bright spots (1–3) in (**c**) are selected for further analysis (white dashed circles). Scale bars, 10 μm. **d** PL spectra of spots 1–3 in (**c**). The emission wavelength and FWHM are 693.98 nm and -0.65 nm for spot 1, 679.09 nm and -1.31 nm for spot 2, and 652.01 nm and -0.79 nm for spot 3. **e** Measured second-order correlation functions $g^{(2)}(\tau)$ using the HBT setup. The $g^{(2)}(0)$ values are $0.229 \pm 0.041$ (spot 1), $0.326 \pm 0.054$ (spot 2), and $0.345 \pm 0.059$ (spot 3). **f** Light in-light out curve at spot 1, exhibiting a saturation power of 22.2 μW and a saturation emission intensity of 67,213 counts/s.

polymethyl methacrylate (PMMA) nanorods. Figure 6a, b show SEM and optical microscope images, respectively, of the exfoliated sheet on 500-nm-high nanorods fabricated using PMMA overdose, resulting in a tensile-strained morphology in BA $n = 2$. Spatial PL mapping was performed at 4 K under pulsed excitation using a 650 nm long-pass filter, comparing near-bandgap excitation ($\lambda_p = 600$ nm) and sub-bandgap

excitation ($\lambda_p = 633$ nm) around the artificially engineered edges (Fig. 6c, d). Five representative spots (1–5) were selected, where the PL spectra exhibited distinct narrow emission peaks ranging from 655.7 to 726.5 nm (Fig. 6e). In addition, the second-order correlation function, $g^{(2)}(\tau)$, measured from spots 2 and 3 showed anti-bunching behavior, with $g^{(2)}(0)$ values of $0.451 \pm 0.045$ and $0.491 \pm 0.067$, respectively

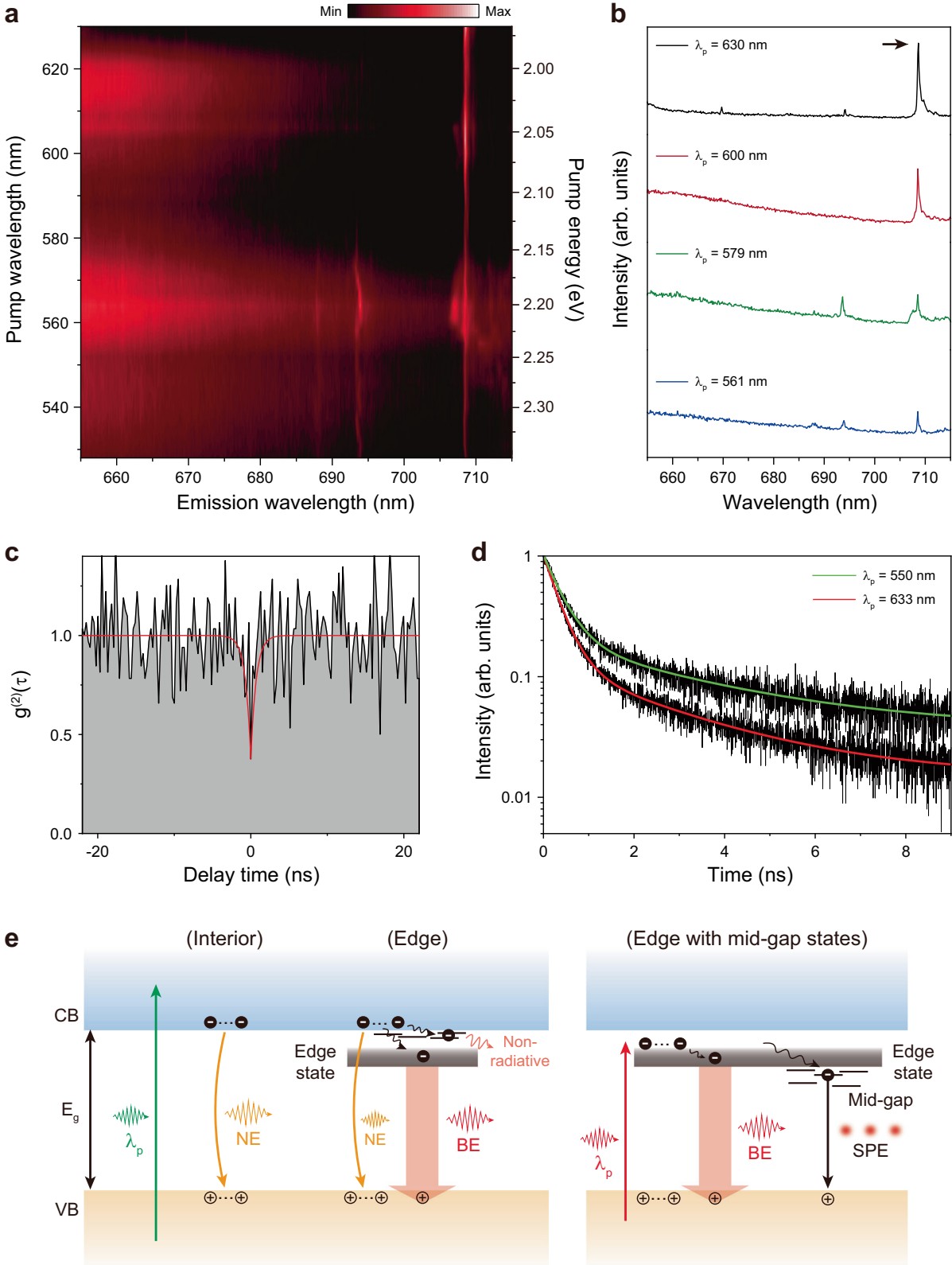

**Fig. 5 | Emission properties of 2D perovskites with different excitation energies. a** PLE map of BA $n$ = 2. The emission wavelength is plotted as a function of the pump wavelength ranging from 528 to 630 nm. **b** PL spectra at pump wavelengths ($λ_p$) of 561, 579, 600, and 630 nm. The intensity of the sharp peak at 708.5 nm (black arrow) increases with an increase in $λ_p$. **c** Measured second-order correlation function $g^{(2)}(τ)$ from the peak indicated by the black arrow in (**b**). The value of $g^{(2)}(0)$ is 0.376 ± 0.167. **d** TRPL measurements at $λ_p$ = 550 nm (green) and 633 nm (red). **e** Proposed model for narrow emission (NE), broad emission (BE), and single-photon emission (SPE) occurring in the interior and edge of 2D perovskites. CB, VB, and $E_g$ indicate conduction band, valence band, and bandgap, respectively. Both above-bandgap (left and middle) and sub-bandgap (right) excitations are illustrated.

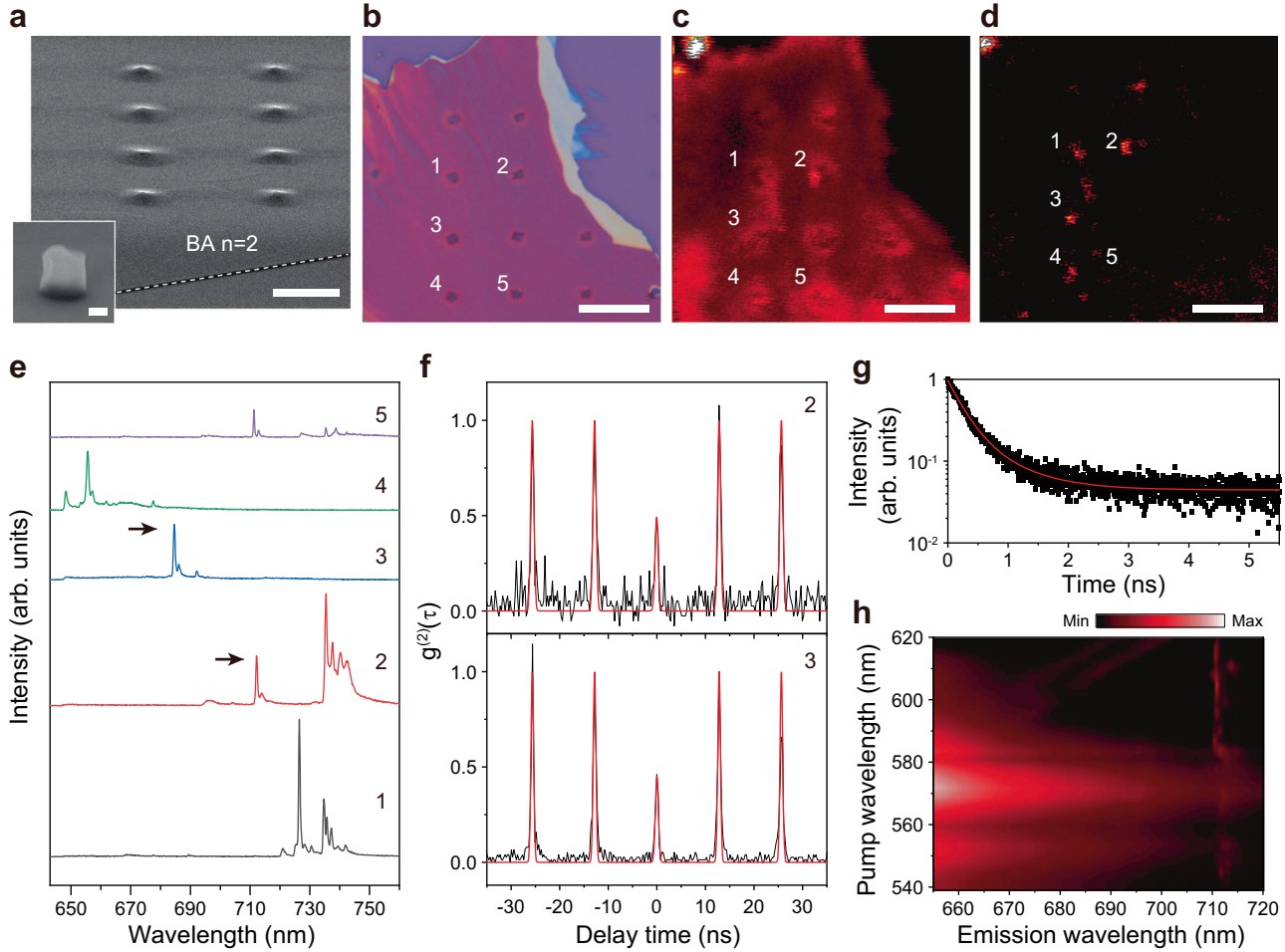

**Fig. 6 | Single-photon emission from BA $n$ = 2 with artificially engineered edges.** **a** SEM image of an exfoliated BA $n$ = 2 single-crystal sheet placed on a SiO$_2$/Si substrate with a PMMA nanorod array. The inset shows a magnified image of the fabricated nanorod. Scale bars, 5 μm and 200 nm (inset). **b** Optical microscope image of the sample. The numbered labels (1–5) indicate the positions investigated. Scale bar, 20 μm. **c, d** Spatial PL intensity maps of the region shown in (**b**), measured at 4 K using a 650 nm long-pass filter under pulsed excitation at $\lambda_p$ = 600 nm (**c**) and 633 nm (**d**). Five localized bright spots (1–5) in (**d**) are selected for further analysis. Scale bars, 10 μm. **e** PL spectra acquired from spots 1–5 in (**d**). **f** Measured second-order correlation functions, g$^{(2)}(\tau)$, for spots 2 and 3, yielding g$^{(2)}$(0) = 0.451 ± 0.045 (spot 2) and 0.491 ± 0.067 (spot 3). **g** TRPL measurement of the emitter at spot 3. **h** PLE map of spot 2, plotted as emission wavelength versus pump wavelength.

(Fig. 6f). Furthermore, TRPL measurements revealed a decay time of 0.696 ± 0.028 ns for the emitter at spot 3 (Fig. 6g). PLE spectroscopy of spot 2 further confirms that the localized emission can be efficiently accessed under sub-bandgap excitation (Fig. 6h).

Spatial PL mapping under lower-energy sub-bandgap excitation reveals strongly localized emitters at the strain-engineered regions, with the localization becoming more pronounced as the pump wavelength is increased from 532 to 633 nm (Fig. 6c, d and Supplementary Fig. 12). Similar behavior was also observed for BA $n$ = 3 samples with artificially engineered edges (Supplementary Fig. 13). Because these strain-induced emitters exhibit the same characteristics as natural edge-state emitters, our results support the idea that tensile strain can generate edge-like defects. Taken together, introducing artificial edges via strain engineering provides a promising route for improved control and optimization of single-photon emitters in 2D perovskites.

In conclusion, we successfully demonstrated 2D perovskite-based single-photon emitters for the first time. These emitters form from deep mid-gap states that are channeled via low-energy edge states through unconventional sub-bandgap excitation. With conventional above-bandgap excitation, the interior region of exfoliated sheets exhibited narrow PL emission, whereas the edge regions displayed broad emission characteristics. Adjusting the pump wavelength

revealed that sub-bandgap excitation induces localized bright spots at the edges of exfoliated sheets of 2D perovskites, particularly in BA $n \geq 2$. These localized bright spots had distinct sharp peaks, which were verified as single-photon emitters by second-order correlation measurements. Exciton dynamics were further investigated through PLE and TRPL measurements, revealing the presence of low-energy edge states within the band that can provide efficient access to deeper mid-gap states responsible for single-photon emission. In particular, 2D perovskite single-photon emitters showed high single-photon purity and brightness accompanied by their rapid decay time. Single-photon emitters were also achieved in artificially engineered edges in exfoliated sheets, which pave an additional avenue for potential emitter-site control.

We summarized the emission energies of single-photon emitters for two compositions (BA $n$ = 2 and BA $n$ = 3) and categorized them into three types (Supplementary Fig. 14). Statistical analysis reveals distinct emission ranges depending on the composition and emitter type. For both BA $n$ = 2 and BA $n$ = 3, for example, single-photon emission predominantly appears in an energy range of ~300–400 meV below the bandgap. These results indicate that, although individual emitter characteristics vary from site to site, selecting the $n$ value and controlling the emission type enable practical control over the

achievable single-photon emission range. We also note that isolated single peaks are not uniformly observed when rastering along the material edges. For future optimization of emission sites and to enable practical implementation, we envision that integrating artificial-edge single-photon emitters with a high-quality optical cavity supporting a single resonant mode could provide reliable single-peak quantum emission.

Overall, 2D perovskites offer distinct advantages, combining solution processability with the favorable properties typically associated with epitaxially grown van der Waals quantum emitters. Furthermore, owing to their intrinsic quantum-well structure, 2D perovskite quantum emitters are free from the monolayer constraint of TMDs, while retaining precise emitter-site control capabilities that are inaccessible to quantum dot-based emitters. Therefore, the unique low-energy edge-state mediated single-photon emission mechanism in 2D perovskites highlights their fundamentally distinct photophysical nature, opening new opportunities for controllable quantum light generation.

## Methods

### Synthesis of 2D perovskites
A slow cooling crystallization method was used to obtain bulk 2D perovskite single crystals. $PbI_2$ (0.4 mmol) and BAI (0.4 mmol) precursors were mixed into an acid mixture containing 0.9 mL of hydroiodic acid (HI, 57 wt % in $H_2O$) and 0.1 mL of hypophosphorous acid ($H_3PO_2$, 50 wt % in $H_2O$) in a glass vial for $(BA)_2PbI_4$ (BA $n = 1$). $PbI_2$ (0.4 mmol), BAI (0.2 mmol), and MAI (0.35 mmol) precursors were mixed into the same acid mixture for $(BA)_2(MA)Pb_2I_7$ (BA $n = 2$). $PbI_2$ (0.6 mmol), BAI (0.2 mmol), and MAI (0.55 mmol) precursors were mixed into the same acid mixture for $(BA)_2(MA)_2Pb_3I_{10}$ (BA $n = 3$). After mixing the precursors and solution, the contents of the sample vial were magnetically stirred and heated to 100 °C in a water bath until materials were completely dissolved and the solution looked clear. The vials were then moved to a Dewar flask water bath at 100 °C to cool down for 72 h until it reached room temperature. The crystals were then retrieved from the solution through vacuum filtration for further use.

### Sample fabrication
2D perovskite single-crystal sheets were isolated from bulk single crystals through mechanical exfoliation using polyvinyl chloride low-adhesion cleanroom tape (blue tape, DN-280). The exfoliated single-crystal sheets were initially transferred onto PDMS using the same tape and subsequently onto a $SiO_2/Si$ substrate via the PDMS stamping method at room temperature.

### Optical measurements
Optical measurements were conducted on the samples at 4 K using a home-built confocal microscope mounted on a helium-flow cryostat (Montana Instruments, s50) (Supplementary Fig. 15). A high-resolution raster scan was performed using two scanning galvanometric mirrors and a 4f confocal alignment system. The PL emission from the sample was collected using a 100× objective lens with a numerical aperture of 0.85 and directed to either a monochromator coupled with a CCD (Princeton Instruments, PIXIS 400 BRX) or to avalanche photodiodes (Excelitas, SPCM AQRH 16). Photon statistics were measured using the HBT setup, wherein photons were detected by two identical avalanche photodiodes coupled with time-tagging electronics (PicoQuant, Pico-Harp 300). The $g^{(2)}(0)$ values were subsequently extracted from well-isolated peaks observed in the samples. For TRPL measurements, the emitters were optically excited using a supercontinuum pulsed laser (NKT Photonics) operating at a repetition rate of 20 MHz. PLE measurements were conducted with a supercontinuum pulsed laser (NKT Photonics) coupled with a SuperK VARIA filter module (NKT Photonics) to selectively tune the pump wavelength.

## Data availability
The data supporting the findings of this study are available within the article and its Supplementary Information. And all data are available from the corresponding authors upon request. Source data are provided with this paper.

## Code availability
The codes used in this work are available from the corresponding authors upon request.

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

## Acknowledgements

H.-G.P. acknowledges support from the National Research Foundation of Korea (NRF) grant funded by the Korean Government (MSIT) (RS-2026-25468766 and RS-2021-NR060087) and the Samsung Science and Technology Foundation (project no. SSTF-BA2401-02). H.-G.P. and J.-P.S. acknowledge support from the NRF grant funded by the Korean Government (MSIT) (RS-2025-02311610). J.-P.S. acknowledges support from the NRF grant funded by the Korean Government (MSIT) (RS-2025-23323094, RS-2025-25460412), the Ministry of Education (RS-2025-25441317), and the Institute for Information and Communications Technology Planning and Evaluation (IITP) (RS-2025-25464252). L.D. acknowledges support from the US Department of Energy (award no. DE-SC0022082). J.Y.P. and P.G. acknowledge support from the Office of Naval Research under Grant N00014-24-1-2045.

## Author contributions

G.N., J.Y.P., L.D., and H.-G.P. designed the experiments. G.N. fabricated the samples, conducted optical measurements, and carried out imaging. J.Y.P. synthesized and characterized the materials. G.N., J.Y.P., J.-P.S., P.G., L.D., and H.-G.P. analyzed the experimental data. All authors contributed to writing and editing the manuscript.

## Competing interests

The authors declare no competing interests.
