## [Transparent Peer Review file · Nature Communications]

Single-photon emission from two-dimensional perovskites channeled through low-energy edge states

Corresponding Author: Professor Hong-Gyu Park

Version 0:

Reviewer comments:

Reviewer #1

(Remarks to the Author)

Edge states of layered halide perovskites have recently been recognized as key factors in enhancing the performance of thin film solar cells, photodetectors, and light emitting diodes, owing to their ability to dissociate excitons into free carriers with lower energy and longer lifetime. In this work, spatially confined single photon emitters were realized at the edges of exfoliated perovskite single crystals through comprehensive spectroscopic measurements. The study is interesting and brings fresh idea on the edge states of 2D halide perovskites. This work is therefore suitable for publication in Nature Communications, if the following concerns can be adequately addressed.

1. The manuscript states that the main features of edge states are their lower energy and longer lifetime compared with bulk states. The authors present PL spectra of $n = 2$ and $n = 3$ samples at 4 K, which differ significantly from previous reports at room temperature. Although PL mapping shows bright edge emission under sub bandgap pumping, are the low temperature edge states physically the same as those at room temperature? Could the extremely low temperature substantially alter the behavior of the edge states?
2. In Figure 3, the authors demonstrate single photon emission at selected edge sites with different emission energies. However, similar emission spots are also observed in the interior region. Hence, the emission appears more likely to be related to point defects. How can the authors clearly separate the contributions from point defects and edge states? Is further experimental or analytical evidence needed to support this distinction?
3. Multiple emission peaks are detected in the perovskite single photon emitters. For a given material, can the energy/wavelength of its single photon emission be predicted? Is it possible to establish a correlation between structure/composition and emission characteristics to guide material design and device applications?
4. The authors artificially created "edge like defects" by covering exfoliated crystals on a SiO_2/Si substrate patterned with PMMA nanorods. The nanorods undoubtedly introduce tensile strain on the crystals. Because the lattice is discontinuous at the edges, the physical environment at the crystal edge is more complex than in the bulk. Why do the authors believe that tensile strain can produce "edge like" defects? Is there any structural or other experimental evidence to support this assumption?
5. The occurrence of edge states in a family of 2D A_2PbBr_4 ($n=1$) should be mentioned in the background. (Adv. Mater. 2025, 37, 2419606)

Reviewer #2

(Remarks to the Author)

In this work, Dou and Park et al introduce a conceptually appealing framework in which 'edge-state' mediated excitation pathways in layered RP perovskites leads to localized single-photon emitters, representing an extension of perovskite photophysics into the quantum-emitter domain. The work's primary strength lies in establishing a connection between material dimensionality, edge electronic structure, and the activation of deep, localized radiative centers that exhibit sub-Poissonian photon statistics. However, the central weaknesses are that the claimed single-photon emission requires more conclusive evidence, as the modest antibunching depth and limited emitter statistics leave open alternative explanations such as background mixing, multiple emitters within the diffraction-limited spot, or blinking-related artifacts, and that the mechanistic interpretation remains largely qualitative, with no explicit identification of the underlying defect structure or atomistic origin. Addressing these issues would substantially elevate the manuscript's rigor and strengthen its suitability for publication at the Nature Communications level. Below are further technical issues for the authors to consider:

1. The reported uncorrected $g(2)(0)$ values ($\sim 0.28\text{--}0.47$) indicate antibunching but remain well above those typical for high-purity SPEs (e.g., $\sim 0.05\text{--}0.15$ for defect centers in hBN, < 0.10 for epitaxial III–V materials). To accurately assess single-photon purity, the manuscript is encouraged to provide background-corrected $g(2)(0)$ values, explicit signal-to-background ratios at each emitter, and ideally power-dependent antibunching measurements to rule out multi-emitter contributions or background-induced artefacts. Residual background emission, whether originating from edge states, trap-related PL, or substrate fluorescence, can artificially suppress or distort the measured dip in $g(2)(0)$, leading to an apparent antibunching signature that does not necessarily reflect the intrinsic single-photon statistics of the emitter itself.
2. The analysis is currently based on a limited number of representative emitters. To establish reproducibility, the authors are encouraged to report distributions of linewidths, brightness, emission wavelength stability, and emitter density along edges, etc. Even a modest statistical set would significantly strengthen the generality of the conclusions.
3. While the edge-state funneling picture is plausible, the manuscript lacks concrete identification of the emitting defect. The authors are encouraged to discuss realistic defect candidates (halide vacancies, reduced-edge octahedra, termination configurations, etc.), ideally supported by DFT levels or relevant literature, and clarify how alternative explanations (e.g., inclusions, local strain pockets, impurities) were excluded.
4. Since narrow-line emission disappears above ~ 60 K (Fig. S6), the authors could include a more realistic assessment of the temperature-limiting mechanisms (e.g., phonon broadening, defect ionization, edge-state delocalization?).
5. Key metrics such as spectral wandering, blinking dynamics, and emission intermittency are not characterized. These are critical for evaluating SPE usability. Time-resolved spectral tracking and intensity autocorrelation analysis over minutes to hours would meaningfully strengthen the case for stable quantum emission.

Reviewer #3

(Remarks to the Author)

The manuscript reports single-photon emission from mid-gap states in $(\text{BA})_2(\text{MA})\text{Pb}_2\text{I}_7$ and $(\text{BA})_2(\text{MA})_2\text{Pb}_3\text{I}_{10}$. The reported synthesis and characterization of these materials are methodologically sound, and claims regarding the lower energy emission are generally compelling based on the evidence provided. My assessment is that this manuscript represents a contribution to the field, particularly toward understanding of the electronic landscape in two-dimensional perovskites, and I would recommend its publication. However, I believe the following concerns should be addressed:

Single-photon emission with regard to practical use in quantum emitters depends on a reliable source of single photons on demand. The $g^{(2)}$ values reported here do meet the criterion for anti-bunching behavior, however, they indicate a relatively high probability of multi-photon emission. The results here represent a characterization of the mid-gap states in the materials studied with interesting results, but these results do not support a claim for practical utility. The manuscript is well-written and pleasant to read, but embellished, and I suggest generally reframing the abstract, introduction, and other sections of the paper to avoid the inference that these materials are of practical use as single-photon emitters.

The second-order correlation measurements in Fig. 3. e., 4. c., and Supplementary Fig. 5. e. exhibit relatively low signal-to-noise ratios, which is unfortunate considering the measurement's importance in establishing the claimed single-photon emission and the quantification of photon purity. Can the reliability of the data be improved, for instance, to the standard of Fig. 5 e.? Were additional pulsed measurements attempted?

Can the authors comment on the general probability and ease of finding a single peak, as opposed to a multitude, when rastering the edges of the materials measured here? Did the artificially engineered edges facilitate isolation of single peaks? Commenting on this could guide future optimization of emission sites and realistically define practical use, if any.

In addition:

What is the photoluminescence lifetime of the mid-gap state emission measured for BA $n=3$ crystals (Fig. 5, Supplementary Fig. 2. and 5.)? How does the lifetime depend on the energetic position of the specific mid-gap state probed, and how does this vary between $n=2$, $n=3$, ... $n=N$ materials?

What is the identity of the emission peak at approximately 694 nm, predominantly visible under above-bandgap excitation (Fig. 4. a, b), and is it significant to the properties of the mid-gap state measured?

Can the authors comment on the thermally activated depopulation of the mid-gap state emission (Supplementary Fig. 6.)? In a recent publication, dark excitons have been reported to localize at the edges of two-dimensional perovskites (Bailey, C. G. et al. *Advanced Energy Materials* **15**, 2501593, 2025). In addition to the edge state, could such fine structure states mediate the population and depopulation of the mid-gap states studied here? Additional temperature-dependent photoluminescence data could address the origins of the thermal quenching and elucidate relaxation pathways relevant to the single-photon emission state.

Version 1:

Reviewer comments:

Reviewer #1

(Remarks to the Author)

The authors have addressed all my concerns. I would like to recommend its publication in Nature Communications now.

Reviewer #2

(Remarks to the Author)

All my concerns have been well addressed. I am supportive of the publication of this work.

Reviewer #3

(Remarks to the Author)

Considering the scope of the work, I believe the concerns regarding the inference of practical use, the SNR of the $g^{(2)}$ measurements, and acquisition of TRPL data have been adequately addressed. Furthermore, the clarifications regarding the isolated single peaks and spectral features are satisfactory. I have no further concerns.

Response to Reviewer #1.

Comment. Edge states of layered halide perovskites have recently been recognized as key factors in enhancing the performance of thin film solar cells, photodetectors, and light emitting diodes, owing to their ability to dissociate excitons into free carriers with lower energy and longer lifetime. In this work, spatially confined single photon emitters were realized at the edges of exfoliated perovskite single crystals through comprehensive spectroscopic measurements. The study is interesting and brings fresh idea on the edge states of 2D halide perovskites. This work is therefore suitable for publication in Nature Communications, if the following concerns can be adequately addressed.

Our response. We thank Reviewer #1 for his/her positive evaluation of the novelty and importance of our work. We are happy to have the opportunity to address the reviewer's important questions and specific suggestions.

Comment 1. The manuscript states that the main features of edge states are their lower energy and longer lifetime compared with bulk states. The authors present PL spectra of $n = 2$ and $n = 3$ samples at 4 K, which differ significantly from previous reports at room temperature. Although PL mapping shows bright edge emission under sub bandgap pumping, are the low temperature edge states physically the same as those at room temperature? Could the extremely low temperature substantially alter the behavior of the edge states?

Our response. We thank the reviewer for raising this insightful question. However, we would like to clarify that we did not state in the manuscript that the lifetime of the edge states is longer than that of the bulk states. In the revised manuscript, we further investigated the temperature dependence of the edge states by performing PL mapping on the same BA $n=2$ flake at 4 K and room temperature (new Supplementary Fig. 2).

Our results suggest that low temperature does not fundamentally change the main features of the edge states. Under various pump wavelengths (532, 580, 600, and 633 nm), broad emission from the flake edges is also enhanced at room temperature, similar to the low-temperature case. Localized emission spots are observed at the edges even at room temperature, although their brightness under 600 nm and 633 nm excitation is weaker than at 4 K. These observations indicate that the edge-state-related emission mechanism is already present at room temperature. In addition, the fact that edge emission occurs over a broad temperature range but becomes more spectrally resolved at low temperature is consistent with a previous report [ACS Nano 13, 1635 (2019)].

To respond to the reviewer's comment, we added new Supplementary Fig. 2 to the revised Supplementary Information. In addition, we added a sentence to the revised manuscript (page 9): "These features observed at 4 K are also present at room temperature (Supplementary Fig. 2), indicating that lowering the temperature does not substantially alter the characteristics of the edge states."

[New Supplementary Fig. 2]

Comment 2. In Figure 3, the authors demonstrate single photon emission at selected edge sites with different emission energies. However, similar emission spots are also observed in the interior region. Hence, the emission appears more likely to be related to point defects. How can the authors clearly separate the contributions from point defects and edge states? Is further

experimental or analytical evidence needed to support this distinction?

Our response. We thank the reviewer’s insightful comment. As the reviewer pointed out, similar emission spots can also be observed in the interior region. Although such interior point-defect single-photon emitters are relatively rare, they can be distinguished from edge-state single-photon emitters based on their spectral and dynamical signatures. A comparison of these emitters, together with the artificial-edge single-photon emitters, is summarized in new Supplementary Fig. 14.

We note that, in contrast to point-defect single-photon emitters, edge-state single-photon emitters occupy a systematically lower-energy band; for BA $n=2$, they are emphasized at $\sim 300\text{--}400$ meV below bandgap energy ($E_g = 2.17$ eV), consistent with our proposed mechanism. In addition, to evaluate emission-wavelength stability, we included time-resolved spectral tracking for both interior point-defect and natural edge-state single-photon emitters (new Supplementary Fig. 9). These results show that the edge-state single-photon emitters are less prone to bleaching and exhibits better spectral stability than the point-defect single-photon emitters.

To respond to the reviewer’s comment, we added new Supplementary Figs. 9 and 14 to the revised Supplementary Information. In addition, we added the following sentences to the revised manuscript (page 14): “While sub-bandgap excitation yielded single-photon emitters at the edges, we rarely observed localized bright spots in the interior that exhibited anti-bunching behavior, with $g^{(2)}(0) = 0.471 \pm 0.047$ (Supplementary Fig. 8). However, the emission-wavelength stability of these interior point-defect single-photon emitters is distinct from that of edge-state emitters. For quantitative evaluation, we performed time-resolved spectral tracking for both interior point-defect and natural edge-state single-photon emitters (Supplementary Fig. 9). The measurements indicate that edge-state single-photon emitters are less prone to bleaching and exhibit better spectral stability than point-defect single-photon emitters.”.

[New Supplementary Fig. 14]

[New Supplementary Fig. 9]

Comment 3. Multiple emission peaks are detected in the perovskite single photon emitters. For a given material, can the energy/wavelength of its single photon emission be predicted? Is it possible to establish a correlation between structure/composition and emission characteristics to guide material design and device applications?

Our response. As the reviewer noted, multiple emission peaks are observed in the perovskite single-photon emitters. As discussed in the original manuscript, the single-photon emitters originate from deep mid-gap states, and thus their exact emission wavelengths cannot be deterministically predicted. However, the emission-wavelength range can be experimentally identified, as shown in new Supplementary Fig. 14a (Comment 2).

Specifically, we summarized the emission energies of single-photon emitters for two compositions (BA $n=2$, $E_g \sim 2.17$ eV; BA $n=3$, $E_g \sim 2.03$ eV), and categorized the emitters into three types such as interior point-defect, natural edge-state, and artificial edge-state single-photon emitters. The statistical analysis reveals distinct emission ranges depending on the composition and type. For example, for both BA $n=2$ and BA $n=3$, single-photon emission predominantly appears in an energy range ~ 300 – 400 meV below E_g . These results indicate that, although individual emitter characteristics vary from site to site, selecting the n value and controlling the emission type enables practical control over the achievable single-photon emission range.

To respond to the reviewer’s comment, we added a paragraph to the revised manuscript (page 21): “We summarized the emission energies of single-photon emitters for two compositions (BA $n=2$ and BA $n=3$) and categorized them into three types (Supplementary Fig. 14). Statistical analysis reveals distinct emission ranges depending on the composition and emitter type. For both BA $n=2$ and BA $n=3$, for example, single-photon emission predominantly appears in an energy range of ~ 300 – 400 meV below the bandgap. These results indicate that, although individual emitter characteristics vary from site to site, selecting the n value and controlling the emission type enable practical control over the achievable single-photon emission range.”

Comment 4. The authors artificially created “edge like defects” by covering exfoliated crystals

on a SiO₂/Si substrate patterned with PMMA nanorods. The nanorods undoubtedly introduce tensile strain on the crystals. Because the lattice is discontinuous at the edges, the physical environment at the crystal edge is more complex than in the bulk. Why do the authors believe that tensile strain can produce “edge like” defects? Is there any structural or other experimental evidence to support this assumption?

Our response. We thank the reviewer for raising this important question. The artificially engineered single-photon emitters formed by tensile strain exhibit the same distinctive characteristics as the natural edge-state single-photon emitters. Since the experimentally observed physical behavior and excitation-dependent signatures are essentially identical, our results support that tensile strain can produce “edge-like” defects.

To clarify this point, we performed additional experiments using an exfoliated BA $n=2$ single-crystal sheet transferred onto a SiO₂/Si substrate patterned with polymethyl methacrylate (PMMA) nanorods (new Fig. 6 and new Supplementary Fig. 12). The measurements reveal several key features. First, spatial PL mapping under lower-energy sub-bandgap excitation produces strongly localized emitters at the strain-engineered regions. This localization becomes increasingly pronounced as the pump wavelength is increased from 532 nm to 633 nm (new Fig. 6c-d and new Supplementary Fig. 12c-f). Second, under deeper sub-bandgap excitation ($\lambda_p = 633$ nm), multiple localized bright spots around the engineered edge-like morphology exhibit distinct narrow emission peaks spanning 655.7–726.5 nm (new Fig. 6e). Corresponding HBT measurements yield $g^{(2)}(0) = 0.451 \pm 0.045$ and 0.491 ± 0.067 (new Fig. 6f). Third, TRPL measurements indicate an emitter decay time of 0.696 ± 0.028 ns (new Fig. 6g). PLE spectroscopy further confirms that the localized emission can be efficiently accessed under sub-bandgap excitation (new Fig. 6h).

Taken together, these optical characteristics closely match those observed for natural edge-state single-photon emitters, supporting our interpretation. The introduction of artificial edges through strain engineering represents a promising approach for additional control and optimization of single-photon emitters in 2D perovskites.

To respond to the reviewer’s question, we have added new Fig. 6 and new Supplementary Fig. 12. We also added the corresponding paragraph to the revised manuscript: “Next, to artificially induce edge-like effects within the interior regions of 2D perovskites, an exfoliated BA $n=2$ single-crystal sheet was transferred onto a SiO₂/Si substrate including patterned polymethyl methacrylate (PMMA) nanorods. Figs. 6a and 6b show SEM and optical microscope images, respectively, of the exfoliated sheet on 500-nm-high nanorods fabricated using PMMA overdose, resulting in a tensile-strained morphology in BA $n=2$. Spatial PL mapping was performed at 4 K under pulsed excitation using a 650 nm long-pass filter, comparing near-bandgap excitation ($\lambda_p = 600$ nm) and sub-bandgap excitation ($\lambda_p = 633$ nm) around the artificially engineered edges (Figs. 6c-d). Five representative spots (1–5) were selected, where the PL spectra exhibited distinct narrow emission peaks ranging from 655.7 to 726.5 nm (Fig. 6e). In addition, the second-order correlation function, $g^{(2)}(\tau)$, measured from spots 2 and 3 showed anti-bunching behavior, with $g^{(2)}(0)$ values of 0.451 ± 0.045 and 0.491 ± 0.067 , respectively (Fig. 6f). Furthermore, TRPL measurements revealed a decay time of 0.696 ± 0.028 ns for the emitter at spot 3 (Fig. 6g). PLE spectroscopy of spot 2 further confirms that the localized emission can be efficiently accessed under sub-bandgap excitation (Fig. 6h).” (page 18) and,

“Spatial PL mapping under lower-energy sub-bandgap excitation reveals strongly localized emitters at the strain-engineered regions, with the localization becoming more pronounced as the pump wavelength is increased from 532 nm to 633 nm (Figs. 6c-d and Supplementary Fig. 12). Similar behavior was also observed for BA $n=3$ samples with artificially engineered edges (Supplementary Fig. 13). Because these strain-induced emitters exhibit the same characteristics as natural edge-state emitters, our results support the idea that tensile strain can generate ‘edge-like’ defects. Taken together, introducing artificial edges via strain engineering provides a promising route for improved control and optimization of single-photon emitters in 2D perovskites.” (page 20).

[New Supplementary Fig. 12]

[New Fig. 6]

Comment 5. The occurrence of edge states in a family of 2D A₂PbBr₄ (n=1) should be mentioned in the background. (Adv. Mater. 2025, 37, 2419606)

Our response. We thank the reviewer for the suggestion. We have cited the recommended work [Adv. Mater. 37, 2419606 (2025)] as Ref. 30 in the revised reference list.

Response to Reviewer #2.

Comment. In this work, Dou and Park et al introduce a conceptually appealing framework in which ‘edge-state’ mediated excitation pathways in layered RP perovskites leads to localized single-photon emitters, representing an extension of perovskite photophysics into the quantum-emitter domain. The work’s primary strength lies in establishing a connection between material dimensionality, edge electronic structure, and the activation of deep, localized radiative centers that exhibit sub-Poissonian photon statistics. However, the central weaknesses are that the claimed single-photon emission requires more conclusive evidence, as the modest antibunching depth and limited emitter statistics leave open alternative explanations such as background mixing, multiple emitters within the diffraction-limited spot, or blinking-related artifacts, and that the mechanistic interpretation remains largely qualitative, with no explicit identification of the underlying defect structure or atomistic origin. Addressing these issues would substantially elevate the manuscript’s rigor and strengthen its suitability for publication at the Nature Communications level. Below are further technical issues for the authors to consider:

Our response. We thank Reviewer #2 for his/her constructive comments and suggestions. We are happy to have the opportunity to address the reviewer’s important questions and specific suggestions.

Comment 1. The reported uncorrected $g^{(2)}(0)$ values (~ 0.28 – 0.47) indicate antibunching but remain well above those typical for high-purity SPEs (e.g., ~ 0.05 – 0.15 for defect centers in hBN, < 0.10 for epitaxial III–V materials). To accurately assess single-photon purity, the manuscript is encouraged to provide background-corrected $g^{(2)}(0)$ values, explicit signal-to-background ratios at each emitter, and ideally power-dependent antibunching measurements to rule out multi-emitter contributions or background-induced artefacts. Residual background emission, whether originating from edge states, trap-related PL, or substrate fluorescence, can artificially suppress or distort the measured dip in $g^{(2)}(0)$, leading to an apparent antibunching signature that does not necessarily reflect the intrinsic single-photon statistics of the emitter itself.

Our response. We thank the reviewer for raising this important point. To improve the reliability of the data, we performed additional pulsed measurements, which increased the signal-to-noise ratio of the second-order correlation function $g^{(2)}(\tau)$ (new Fig. 4). Compared with CW excitation, pulsed excitation suppresses residual background emission, allowing the intrinsic single-photon statistics of the emitter to be measured more reliably and yielding improved $g^{(2)}(0)$ values. Representative values are $g^{(2)}(0) = 0.229 \pm 0.041$, 0.326 ± 0.054 , and 0.345 ± 0.059 for spots 1, 2, and 3, respectively, on the same BA $n=2$ flake (new Fig. 4e).

We also performed power-dependent $g^{(2)}(\tau)$ measurements on a single-photon emitter under pulsed excitation (new Supplementary Fig. 7). The observation that $g^{(2)}(0)$ improves at lower pump power ($g^{(2)}(0) \sim 0.20$ at $20 \mu\text{W}$ and ~ 0.23 at $40 \mu\text{W}$) supports that the anti-bunching is dominated by a single emitter rather than being an artifact caused by background noise. Although the single-photon purity in 2D perovskites is currently lower than that of established platforms such as h-BN or epitaxial III-V materials, we emphasize that developing single-photon emitters in non-epitaxially grown materials is pivotal for realizing practical, low-cost, and scalable quantum light sources.

To respond to the reviewer’s comment, we added new Fig. 4 and a corresponding paragraph in the revised manuscript (page 12): “Next, we performed sub-bandgap pulsed excitation measurements on BA $n=2$ at 4 K. Compared with CW excitation, pulsed excitation suppresses residual background emission, allowing the intrinsic single-photon statistics of the emitter to be measured more reliably and yielding improved $g^{(2)}(0)$ values. Fig. 4a shows an optical microscope image of the exfoliated BA $n=2$ single-crystal sheet used for these measurements. Similar to Figs. 2 and 3, spatial PL mapping with a 650 nm long-pass filter revealed increased intensity in the edge region under near-bandgap excitation ($\lambda_p = 600$ nm), whereas sub-bandgap excitation ($\lambda_p = 633$ nm) led to localized bright spots at the edges (Figs. 4b and c). We selected three representative spots (1–3) and observed isolated narrow emission lines with FWHM of ~ 0.65 – 1.31 nm (Fig. 4d). HBT measurements under pulsed excitation yielded clear photon anti-bunching for all three spots, with $g^{(2)}(0) = 0.229 \pm 0.041$ (spot 1), 0.326 ± 0.054 (spot 2), and 0.345 ± 0.059 (spot 3) (Fig. 4e). The measurement of the integrated PL intensity as a function of the pump power exhibited saturation behavior, with a saturation emission intensity of 67,213 counts/s at a pump power of 22.2 μ W (Fig. 4f).”

In addition, we added new Supplementary Fig. 7 and two sentences to the revised manuscript (page 14): “We also measured power-dependent $g^{(2)}(\tau)$ on the same emitter (spot 1 in Fig. 4c) and found that $g^{(2)}(0)$ improves at a lower pump power ($g^{(2)}(0) \sim 0.20$ at 20 μ W and ~ 0.23 at 40 μ W) (Supplementary Fig. 7). This result supports the fact that the observed anti-bunching is dominated by a single emitter, rather than being an artifact arising from background noise.”

While the $g^{(2)}(0)$ values in Fig. 3e and old Fig. 4c exhibit relatively low signal-to-noise ratios, as the reviewer noted, these panels are primarily included to highlight the main observations—the localized bright spots at the edges (Fig. 3e) and the PLE measurements of single-photon emitters (old Fig. 4c).

[New Fig. 4]

[New Supplementary Fig. 7]

Comment 2. The analysis is currently based on a limited number of representative emitters. To establish reproducibility, the authors are encouraged to report distributions of linewidths, brightness, emission wavelength stability, and emitter density along edges, etc. Even a modest statistical set would significantly strengthen the generality of the conclusions.

Our response. We thank the reviewer for raising this important point. As suggested by the reviewer, we measured a larger number of emitters (new Fig. 4 and new Fig. 6) and added reproducibility and statistical analyses (new Supplementary Fig. 14). As shown in new Supplementary Fig. 14, we summarized the emission wavelength/energy distributions, linewidth distributions, and the corresponding $g^{(2)}(0)$ values for three types of single-photon emitters (interior point-defect, natural edge-state, and artificial edge-state single-photon emitters) across two 2D perovskite compositions (BA $n=2$ and BA $n=3$). In addition, to evaluate emission wavelength stability, we included time-resolved spectral tracking for both interior point-defect single-photon emitters and natural edge-state single-photon emitters (new Supplementary Fig. 9).

Regarding emitter density along edges, we agree with the reviewer that this is also an important parameter. However, it strongly depends on the sample and the measurement conditions. Therefore, rather than reporting a value, we present PL maps acquired under various pump wavelengths in the main and supplementary figures.

To respond to the reviewer’s comment, we added new Supplementary Figs. 9 and 14 to the revised Supplementary Information. Also, we added a paragraph to the revised manuscript (page 21): “We summarized the emission energies of single-photon emitters for two compositions (BA $n=2$ and BA $n=3$) and categorized them into three types (Supplementary Fig. 14). Statistical analysis reveals distinct emission ranges depending on the composition and emitter type. For both BA $n=2$ and BA $n=3$, for example, single-photon emission predominantly appears in an energy range of $\sim 300\text{--}400$ meV below the bandgap. These results indicate that, although individual emitter characteristics vary from site to site, selecting the n value and controlling the emission type enable practical control over the achievable single-photon emission range.”

[New Supplementary Fig. 14]

[New Supplementary Fig. 9]

Comment 3. While the edge-state funneling picture is plausible, the manuscript lacks concrete identification of the emitting defect. The authors are encouraged to discuss realistic defect candidates (halide vacancies, reduced-edge octahedra, termination configurations, etc.), ideally supported by DFT levels or relevant literature, and clarify how alternative explanations (e.g., inclusions, local strain pockets, impurities) were excluded.

Our response. We thank the reviewer for raising this point. Identifying the exact origin of single-photon emitters in halide perovskite materials remains experimentally challenging because these materials are intrinsically soft and fragile. In our previous attempts, we used STEM to probe the emitters, but the samples degraded rapidly under the electron beam, preventing reliable defect identification. Very recently, a new TEM technique was developed to image defects in halide perovskite materials [Nature 647, 364 (2025)]; however, this approach requires sophisticated sample preparation and advanced data analysis. We plan to establish new collaborations with TEM experts in the near future to investigate the structural origin of the single-photon emitters in more detail. Even if sufficient structural information can be obtained, unambiguously assigning an emission peak to a specific defect site is still difficult at the present stage. We thus believe that disclosing the exciting observation of single-photon emission in 2D perovskite materials in a timely fashion will benefit the community and help attract additional imaging and theory experts to study this type of materials.

Instead, as suggested by the reviewer, we have added a discussion of realistic defect candidates by outlining possible explanations for edge states based on relevant literature. First, edge-termination-induced reconstruction (i.e., “dangling/relaxed” terminal PbX_6 octahedra) can generate reduced-energy edge-localized states, as predicted by first-principles calculations [Adv. Mater. 34, 2201666 (2022)]. Alternatively, the edges in exfoliated single crystals can be described as layer-edge states stabilized by internal electric fields arising from dipole alignment and termination-dependent electrostatics [Nano Lett. 21, 182 (2021)]. Together, these mechanisms support interpreting our emitters as originating from low-energy “edge states” that are unique to 2D perovskites, rather than from generic bulk point defects.

To address the reviewer’s comment, we added one sentence to the revised manuscript (page 5): “which are attributed to edge-termination-induced reconstruction or termination-dependent electrostatics^{29,31}”.

Comment 4. Since narrow-line emission disappears above ~ 60 K (Fig. S6), the authors could include a more realistic assessment of the temperature-limiting mechanisms (e.g., phonon broadening, defect ionization, edge-state delocalization?).

Our response. We thank the reviewer for raising this point regarding the temperature-limit mechanisms. To provide a more realistic assessment, we performed temperature-dependent TRPL measurements on an edge-state emitter in BA $n=2$ at 4, 10, 20, 40, and 60 K, and analyzed the spectral characteristics, including the emission linewidth (FWHM) and the integrated PL intensity (new Supplementary Fig. 11). From these measurements, we extracted the temperature dependence of the PL lifetime.

The results reveal several key features. First, the linewidth (FWHM) increases from 0.8 nm at 10 K to 5.5 nm at 60 K, consistent with enhanced exciton-phonon interactions and increased spectral fluctuations at elevated temperatures. Second, the integrated PL intensity decreases rapidly above 20 K, and the lifetime decreases from ~ 6 ns near 10 K to ~ 2 ns at 60 K, indicating that nonradiative pathways become increasingly competitive as temperature rises. Third, the non-monotonic lifetime trend, an increase from 4 K to 10 K followed by a decrease, suggests a crossover in the dominant decay pathways. Overall, these results indicate that the practical temperature limit is primarily set by phonon-driven loss of spectral selectivity together with activated competition from nonradiative/escape channels, rather than a single mechanism alone.

To respond to the reviewer's comment, we added new Supplementary Fig. 11 and the following sentences to the revised manuscript (page 17): “Furthermore, we performed temperature-dependent TRPL measurements on an edge-state emitter in BA $n=2$ at 4, 10, 20, 40, and 60 K, and analyzed the spectral characteristics, including the emission linewidth (FWHM) and the integrated PL intensity (Supplementary Fig. 11). We also extracted the temperature dependence of the PL lifetime. A key feature of these measurements is that the practical temperature limit is primarily set by phonon-driven loss of spectral selectivity together with activated competition from nonradiative/escape channels, rather than a single mechanism alone.”

[New Supplementary Fig. 11]

Comment 5. Key metrics such as spectral wandering, blinking dynamics, and emission intermittency are not characterized. These are critical for evaluating SPE usability. Time-resolved spectral tracking and intensity autocorrelation analysis over minutes to hours would meaningfully strengthen the case for stable quantum emission.

Our response. We thank the reviewer for raising this important point regarding emission stability. As suggested by the reviewer, we performed time-resolved spectral tracking (new Supplementary Fig. 9), in which PL spectra are acquired every 10 s for up to 5 min for both a point-defect single-photon emitter (~663 nm) and an edge-state single-photon emitter (~695 nm). These results show that the edge-state single-photon emitter is less prone to bleaching and exhibits better spectral stability than the point-defect single-photon emitter.

[New Supplementary Fig. 9]

To address the reviewer’s comment, we added new Supplementary Fig. 9 and the following sentences to the revised manuscript (page 14): “While sub-bandgap excitation yielded single-photon emitters at the edges, we rarely observed localized bright spots in the interior that exhibited anti-bunching behavior, with $g^{(2)}(0) = 0.471 \pm 0.047$ (Supplementary Fig. 8). However, the emission-wavelength stability of these interior point-defect single-photon emitters is distinct from that of edge-state emitters. For quantitative evaluation, we performed time-resolved spectral tracking for both interior point-defect and natural edge-state single-photon emitters (Supplementary Fig. 9). The measurements indicate that edge-state single-photon emitters are less prone to bleaching and exhibit better spectral stability than point-defect single-photon emitters.”

Response to Reviewer #3.

Comment. The manuscript reports single-photon emission from mid-gap states in $(\text{BA})_2(\text{MA})\text{Pb}_2\text{I}_7$ and $(\text{BA})_2(\text{MA})_2\text{Pb}_3\text{I}_{10}$. The reported synthesis and characterization of these materials are methodologically sound, and claims regarding the lower energy emission are generally compelling based on the evidence provided. My assessment is that this manuscript represents a contribution to the field, particularly toward understanding of the electronic landscape in two-dimensional perovskites, and I would recommend its publication. However, I believe the following concerns should be addressed:

Our response. We thank Reviewer #3 for his/her positive evaluation of the novelty and importance of our work. We are happy to have the opportunity to address the reviewer's important questions and specific suggestions.

Comment 1. Single-photon emission with regard to practical use in quantum emitters depends on a reliable source of single photons on demand. The $g^{(2)}$ values reported here do meet the criterion for anti-bunching behavior, however, they indicate a relatively high probability of multi-photon emission. The results here represent a characterization of the mid-gap states in the materials studied with interesting results, but these results do not support a claim for practical utility. The manuscript is well-written and pleasant to read, but embellished, and I suggest generally reframing the abstract, introduction, and other sections of the paper to avoid the inference that these materials are of practical use as single-photon emitters.

Our response. We thank the reviewer for the positive comments on our results. We also appreciate the reviewer's point regarding the practical use of single-photon emitters. While our manuscript was not intended to claim immediate practical applicability, we recognize that some statements could be interpreted in that way. As suggested by the reviewer, we have therefore revised and toned down several expressions in the Abstract, Introduction, and other sections to avoid potential overstatement and unnecessary confusion. For example, we removed the following sentences in the revised manuscript (page 21): "Beyond these advantages which directly address the limitations of existing quantum emitter materials," and "Our demonstration of 2D perovskite single-photon emitters, combined with their exceptional chemical tunability and structural versatility, establishes them as a promising and scalable candidate for developing efficient quantum light sources for emerging quantum technologies."

Comment 2. The second-order correlation measurements in Fig. 3. e., 4. c., and Supplementary Fig. 5. e. exhibit relatively low signal-to-noise ratios, which is unfortunate considering the measurement's importance in establishing the claimed single-photon emission and the quantification of photon purity. Can the reliability of the data be improved, for instance, to the standard of Fig. 5 e.? Were additional pulsed measurements attempted?

Our response. We thank the reviewer for raising this important point. To improve the reliability of the data, we performed additional pulsed measurements, which increased the signal-to-noise ratio of the second-order correlation function $g^{(2)}(\tau)$ (new Fig. 4). Compared with CW excitation, pulsed excitation suppresses residual background emission, allowing the intrinsic single-photon statistics of the emitter to be measured more reliably and yielding improved $g^{(2)}(0)$ values. Representative values are $g^{(2)}(0) = 0.229 \pm 0.041$, 0.326 ± 0.054 , and 0.345 ± 0.059 for

spots 1, 2, and 3, respectively, on the same BA $n=2$ flake (new Fig. 4e).

While the $g^{(2)}(0)$ values in Fig. 3e and old Fig. 4c exhibit relatively low signal-to-noise ratios, as the reviewer noted, these panels are primarily included to highlight the main observations—the localized bright spots at the edges (Fig. 3e) and the PLE measurements of single-photon emitters (old Fig. 4c).

To respond to the reviewer's comment, we added new Fig. 4 and a corresponding paragraph to the revised manuscript (page 12): “Next, we performed sub-bandgap pulsed excitation measurements on BA $n=2$ at 4 K. Compared with CW excitation, pulsed excitation suppresses residual background emission, allowing the intrinsic single-photon statistics of the emitter to be measured more reliably and yielding improved $g^{(2)}(0)$ values. Fig. 4a shows an optical microscope image of the exfoliated BA $n=2$ single-crystal sheet used for these measurements. Similar to Figs. 2 and 3, spatial PL mapping with a 650 nm long-pass filter revealed increased intensity in the edge region under near-bandgap excitation ($\lambda_p = 600$ nm), whereas sub-bandgap excitation ($\lambda_p = 633$ nm) led to localized bright spots at the edges (Figs. 4b and c). We selected three representative spots (1–3) and observed isolated narrow emission lines with FWHM of ~ 0.65 – 1.31 nm (Fig. 4d). HBT measurements under pulsed excitation yielded clear photon anti-bunching for all three spots, with $g^{(2)}(0) = 0.229 \pm 0.041$ (spot 1), 0.326 ± 0.054 (spot 2), and 0.345 ± 0.059 (spot 3) (Fig. 4e). The measurement of the integrated PL intensity as a function of the pump power exhibited saturation behavior, with a saturation emission intensity of 67,213 counts/s at a pump power of 22.2 μ W (Fig. 4f).”.

[New Fig. 4]

Comment 3. Can the authors comment on the general probability and ease of finding a single peak, as opposed to a multitude, when rastering the edges of the materials measured here? Did the artificially engineered edges facilitate isolation of single peaks? Commenting on this could guide future optimization of emission sites and realistically define practical use, if any.

Our response. We thank the reviewer for raising this question. In our measurements, isolated single peaks are not uniformly observed when rastering along the edges of the materials. For the artificially engineered edges, because they exhibit emission characteristics consistent with those of natural edges (as discussed in Comment 4 of Reviewer 1), isolated single peaks can

similarly be observed. However, artificial-edge emitters are found only at selected localized sites, rather than uniformly along the entire edge (new Fig. 6). For future optimization of emission sites and to realistically define practical use, we envision that combining artificial-edge single-photon emitters with a high-Q optical cavity that supports a single resonant mode could enable reliable single-peak quantum emission.

To respond to the reviewer's comment, we added two sentences to the revised manuscript (page 21): “We also note that isolated single peaks are not uniformly observed when rastering along the material edges. For future optimization of emission sites and to enable practical implementation, we envision that integrating artificial-edge single-photon emitters with a high-quality optical cavity supporting a single resonant mode could provide reliable single-peak quantum emission.”

Comment 4. What is the photoluminescence lifetime of the mid-gap state emission measured for BA $n=3$ crystals (Fig. 5, Supplementary Fig. 2. and 5.)? How does the lifetime depend on the energetic position of the specific mid-gap state probed, and how does this vary between $n=2$, $n=3$, ... $n=N$ materials?

Our response. We thank the reviewer for raising this point. We performed TRPL measurements on a BA $n=3$ emitter at 4 K to obtain an accurate PL lifetime for the mid-gap state emission. The measured PL lifetime is $\tau = 1.729 \pm 0.032$ ns (new Supplementary Fig. 6). For comparison, the representative lifetime for BA $n=2$ is ~ 0.87 ns (Fig. 3).

At present, we only have TRPL datasets for BA $n=2$ and $n=3$. Thus, we cannot establish how the lifetime varies among $n = 2, 3, \dots, N$ materials. Additional material-dependent studies may be addressed in future work.

To address the reviewer's comment, we added new Supplementary Fig. 6 and the following sentence to the revised manuscript (page 12): “In addition, the PL lifetime of BA $n=3$ was measured to be longer ($\tau = 1.729 \pm 0.032$ ns) than that of BA $n=2$ (Supplementary Fig. 6).”

Comment 5. What is the identity of the emission peak at approximately 694 nm, predominantly visible under above-bandgap excitation (Fig. 4. a, b), and is it significant to the properties of the mid-gap state measured?

Our response. We thank the reviewer for this careful observation. In our excitation-energy-dependent interpretation, multiple recombination pathways exist at the edge (e.g., band-edge emission, edge-state broad emission, and shallow-trap emission). The emission peak at ~694 nm can appear from one of these channels when carriers are generated under above-bandgap excitation. On the other hand, the $g^{(2)}(\tau)$ and TRPL properties of the mid-gap state discussed in Fig. 4 were assessed by spectrally isolating the targeted sharp peak using appropriate filtering. Thus, the presence of the ~694 nm peak does not affect the measured properties of the mid-gap emitter.

Comment 6. Can the authors comment on the thermally activated depopulation of the mid-gap state emission (Supplementary Fig. 6.)? In a recent publication, dark excitons have been reported to localize at the edges of two-dimensional perovskites (Bailey, C. G. et al. *Advanced Energy Materials* **15**, 2501593, 2025). In addition to the edge state, could such fine structure states mediate the population and depopulation of the mid-gap states studied here? Additional temperature-dependent photoluminescence data could address the origins of the thermal quenching and elucidate relaxation pathways relevant to the single-photon emission state.

Our response. We thank the reviewer for raising this point regarding the temperature-limit mechanisms. To provide a more realistic assessment, we performed temperature-dependent

TRPL measurements on an edge-state emitter in BA $n=2$ at 4, 10, 20, 40, and 60 K, and analyzed the spectral characteristics, including the emission linewidth (FWHM) and the integrated PL intensity (new Supplementary Fig. 11). From these measurements, we extracted the temperature dependence of the PL lifetime.

The results reveal several key features. First, the linewidth (FWHM) increases from 0.8 nm at 10 K to 5.5 nm at 60 K, consistent with enhanced exciton-phonon interactions and increased spectral fluctuations at elevated temperatures. Second, the integrated PL intensity decreases rapidly above 20 K, and the lifetime decreases from ~ 6 ns near 10 K to ~ 2 ns at 60 K, indicating that nonradiative pathways become increasingly competitive as temperature rises. Third, the non-monotonic lifetime trend, an increase from 4 K to 10 K followed by a decrease, suggests a crossover in the dominant decay pathways. Overall, these results indicate that the practical temperature limit is primarily set by phonon-driven loss of spectral selectivity together with activated competition from nonradiative/escape channels, rather than a single mechanism alone.

The reviewer's suggestion that dark-exciton fine-structure states may mediate the population and depopulation of the mid-gap states is intriguing and represents a possible pathway. However, since our observations can be consistently interpreted within the edge-state model, we did not further investigate this mechanism in the present study.

To respond to the reviewer's comment, we added new Supplementary Fig. 11 and the following sentences to the revised manuscript (page 17): "Furthermore, we performed temperature-dependent TRPL measurements on an edge-state emitter in BA $n=2$ at 4, 10, 20, 40, and 60 K, and analyzed the spectral characteristics, including the emission linewidth (FWHM) and the integrated PL intensity (Supplementary Fig. 11). We also extracted the temperature dependence of the PL lifetime. A key feature of these measurements is that the practical temperature limit is primarily set by phonon-driven loss of spectral selectivity together with activated competition from nonradiative/escape channels, rather than a single mechanism alone."

[New Supplementary Fig. 11]

Response to Reviewer #1.

Comment. The authors have addressed all my concerns. I would like to recommend its publication in Nature Communications now.

Our response. We thank Reviewer #1 for his/her positive evaluation of our work and the explicit recommendation for publication.

Response to Reviewer #2.

Comment. All my concerns have been well addressed. I am supportive of the publication of this work.

Our response. We thank Reviewer #2 for his/her positive evaluation of our work and the explicit recommendation for publication.

Response to Reviewer #3.

Comment. Considering the scope of the work, I believe the concerns regarding the inference of practical use, the SNR of the $g(2)$ measurements, and acquisition of TRPL data have been adequately addressed. Furthermore, the clarifications regarding the isolated single peaks and spectral features are satisfactory. I have no further concerns.

Our response. We thank Reviewer #3 for his/her positive evaluation of our work and the explicit recommendation for publication.